# An analysis of 45 large-scale wastewater sites in England to estimate SARS-CoV-2 community prevalence

Mario Morvan [1,2,14], Anna Lo Jacomo[1,3,14], Celia Souque[1,4], Matthew J. Wade [1,5], Till Hoffmann [1,6], Koen Pouwels [7,8], Chris Lilley[1], Andrew C. Singer [9], Jonathan Porter [10], Nicholas P. Evens[10], David I. Walker[11], Joshua T. Bunce[1,5,12], Andrew Engeli[1], Jasmine Grimsley[1], Kathleen M. O'Reilly [1,13,14✉] & Leon Danon[1,3,14]

Accurate surveillance of the COVID-19 pandemic can be weakened by under-reporting of cases, particularly due to asymptomatic or pre-symptomatic infections, resulting in bias. Quantification of SARS-CoV-2 RNA in wastewater can be used to infer infection prevalence, but uncertainty in sensitivity and considerable variability has meant that accurate measurement remains elusive. Here, we use data from 45 sewage sites in England, covering 31% of the population, and estimate SARS-CoV-2 prevalence to within 1.1% of estimates from representative prevalence surveys (with 95% confidence). Using machine learning and phenomenological models, we show that differences between sampled sites, particularly the wastewater flow rate, influence prevalence estimation and require careful interpretation. We find that SARS-CoV-2 signals in wastewater appear 4–5 days earlier in comparison to clinical testing data but are coincident with prevalence surveys suggesting that wastewater surveillance can be a leading indicator for symptomatic viral infections. Surveillance for viruses in wastewater complements and strengthens clinical surveillance, with significant implications for public health.

[1] Data, Analytics, and Surveillance Group, UK Health Security Agency (Formerly part of the Joint Biosecurity Centre, Department of Health and Social Care), London SW1P 3JR, UK. [2] Department of Physics and Astronomy, University College London, London WC1E 6BT, UK. [3] Department of Engineering Mathematics, Ada Lovelace Building, University Walk, Bristol BS8 1TW, UK. [4] Department of Zoology, University of Oxford, Oxford OX1 3SZ, UK. [5] School of Engineering, Newcastle University, Newcastle-upon-Tyne NE1 7RU, UK. [6] Department of Mathematics, Imperial College London, London SW7 2AZ, UK. [7] NIHR Health Protection Research Unit in Healthcare Associated Infections and Antimicrobial Resistance at University of Oxford in partnership with Public Health England, Oxford, UK. [8] Health Economics Research Centre, Nuffield Department of Population Health, University of Oxford, Oxford, UK. [9] UK Centre for Ecology & Hydrology, Wallingford OX10 8BB, UK. [10] Environment Agency, National Monitoring, Starcross, Exeter EX6 8FD, UK. [11] Centre for Environment, Fisheries and Aquaculture Science, Weymouth DT4 8UB, UK. [12] Department for Environment, Food and Rural Affairs, London SW1P 4DF, UK. [13] Centre for Mathematical Modelling of Infectious Diseases & Faculty of Epidemiology and Population Health, London School of Hygiene and Tropical Medicine, London WC1E 7HT, UK. [14] These authors contributed equally: Mario Morvan, Anna Lo Jacomo, Kathleen M. O'Reilly, Leon Danon. ✉email: Kathleen.oreilly@lshtm.ac.uk

Estimates of SARS-CoV-2 infection prevalence are essential to understand COVID-19 disease burden and the impact of public health interventions[1–3]. The sensitivity of surveillance for COVID-19 varies for several epidemiological, administrative, political and financial reasons, meaning that reported cases are likely to be an underestimate of actual cases[4–7]. Further, only a proportion of infections result in symptomatic disease; estimates range considerably across studies but in a recent meta-analysis that aimed to account for potential biases in reporting an average of 64.9% (95% CI 60.1–69.3%) was estimated[8], so there is a disconnect between infections that result in transmission and reported cases via clinical surveillance. To monitor the trajectory of the COVID-19 pandemic and reduce the impact of bias due to any single source of information, it is essential to have multiple measures of prevalence with well-understood sources of bias and uncertainty.

Based on its proven success applied to other infectious diseases and markers of human health and behaviour[9,10], wastewater (WW) surveillance for SARS-CoV-2 has been established in many countries, including England, since the start of the pandemic[11,12]. Early studies indicated that fragments of RNA corresponding to the SARS-CoV-2 viral genome were detectible in WW[13], and in quantities that some quantitative measure could be established. However, laboratory protocols required refinement to establish a method that provides consistent measures of RNA with sufficient sensitivity and to be used at scale. Quantitative reverse-transcriptase PCR (RT-qPCR) of target genes of the virus genome is now routinely performed on concentrated samples from wastewater.

A challenge inherent to WW surveillance is the potential impact of environmental and biochemical attributes on the detection and quantification of the virus concentration. In England and other countries that utilise combined sewer networks to transport sewage, the wastewater inflowing at the sewage treatment works (STWs) typically comprises a combination of raw sewage, household effluent (e.g. from washing and cleaning), agricultural run-off, rainwater/snow melt, and trade waste from industry[14]. The percentage volume of human-derived excreta likely to contain virus RNA (i.e., urine, faeces, nasal discharge, sputum, blood)[15] in the collected wastewater sample is likely to vary because of additional inflow detailed above, which will dilute and may degrade the target analyte concentration, reducing the sensitivity of lab assays[16]. In order to overcome these challenges and infer an estimate of prevalence from WW samples, additional data and statistical models can be used, and validation of model outputs using reliable estimates of prevalence is critical. However, the associated biases in clinical surveillance and the impact of uncertainties associated with environmental monitoring of viruses in sewer networks present significant challenges when considering prevalence estimation using WW measurements.

Back-calculating or estimating the quantity of chemical compounds (e.g., licit and illicit pharmaceuticals) or stressors (e.g., pathogens) by targeting indicative analytes present in WW is a common feature of wastewater-based epidemiology (WBE). For example, WBE has been applied successfully in estimating illegal drug consumption[9], the degree of antibiotic resistance in a population[17], among other applications. While most studies have used WW to track disease trends (i.e. increase/decrease), a number of studies have attempted to directly quantify prevalence from SARS-CoV-2 measurements, along with biological and hydrological parameters[7,18,19]. Broadly, studies using back-calculation for SARS-CoV-2 generally consider that disease prevalence is equal to the load of RNA in the sample, divided by the load of RNA produced by one infected person[19]. The underlying assumptions are that viral RNA is released proportionally to wastewater and perfectly mixed in the sewers, and that there are

no significant losses of virus RNA in the network that lead to a decrease in measurement representativeness. Variations of this hypothesis have been suggested to account for additional 'signal loss' factors, for example decay, flow dilution, and temporal shedding patterns in the population[19].

The Office for National Statistics (ONS) Covid-19 Infection Survey (CIS) was established in the UK early in the pandemic to assess the prevalence of individuals in the community testing positive for SARS-CoV-2 (otherwise known as "positivity") through nasopharyngeal sampling of individuals living in randomly selected private households from the UK[3]. This survey has been essential to understand the dynamics of SARS-CoV-2 by estimating community positivity rates, and further to estimate these rates at regional and sub-regional scales. The wide availability of WW samples from July 2020 in England and sub-regional positivity estimates from the CIS provides a unique dataset to investigate and validate WW as a reliable estimate of prevalence to support public health actions. In this study, we estimate the prevalence of SARS-CoV-2 infection in the community and establish what additional data and analyses are required to have accurate and robust estimates of prevalence from WW.

## Results

We analysed data collected between July 2020 and March 2021 from 45 sewage treatment works (STWs) across England (Fig. 1) covering an estimated 31% of the population. For each site, an average of four samples were collected per week, by either grab (46%) or composite (54%) sampling. Additional metadata were collected on inorganics and other wastewater characteristics (see Supplementary Table S1 for further details).

Translation of raw WW data to prevalence estimates are illustrated using a phenomenological model that considers infection prevalence, shedding and stool generation, and the volume of water in the sewage column (see the "Methods" section). The assumptions of the model results in a linear relationship between prevalence and RNA concentrations. Sensitivity analysis illustrated that viral concentration in stool is the largest source of uncertainty in this approach (Fig. S2). Using average values of the shedding rate from clinical studies[20] gives a relatively good fit with observations from wastewater and CIS data in terms of average magnitude, but with high variability across individual samples (Fig. 2A). However, that variability is commensurate with the uncertainty in the appropriate hydrological and biological values used for the calculation. Comparing these model estimates to data indicates that for more than half of the CIS sub-regions in the study (60%) the model assumptions illustrate a valid relationship with the data (more than 60% of sample points fall within the 50% confidence interval of the model). Sites showing a poorer fit, have either relatively low (28%) or relatively high (12%) concentrations per positivity rate (Fig. 2B and C). Lower than expected concentrations could be caused by unusually high per capita flow rates (such as groundwater infiltration), or degradation of RNA during transit due to physical or chemical characteristics of the network (such as numerous pumping stations, or consistently atypical pH). The method of sample collection, together with limited homogenisation of the 'sewage parcel', could also lead to unrepresentative (either low or high) concentrations, or indeed unaccounted sewage discharge could also affect measurement[21]. Including an additional factor to account for degradation might provide a better model assumption for sites showing relatively low concentrations (e.g. Sub-region B in Fig. 2A). In some sub-regions, the relationship between concentrations and prevalence is not well explained by the (static) linear model. A possible reason may be interactions between

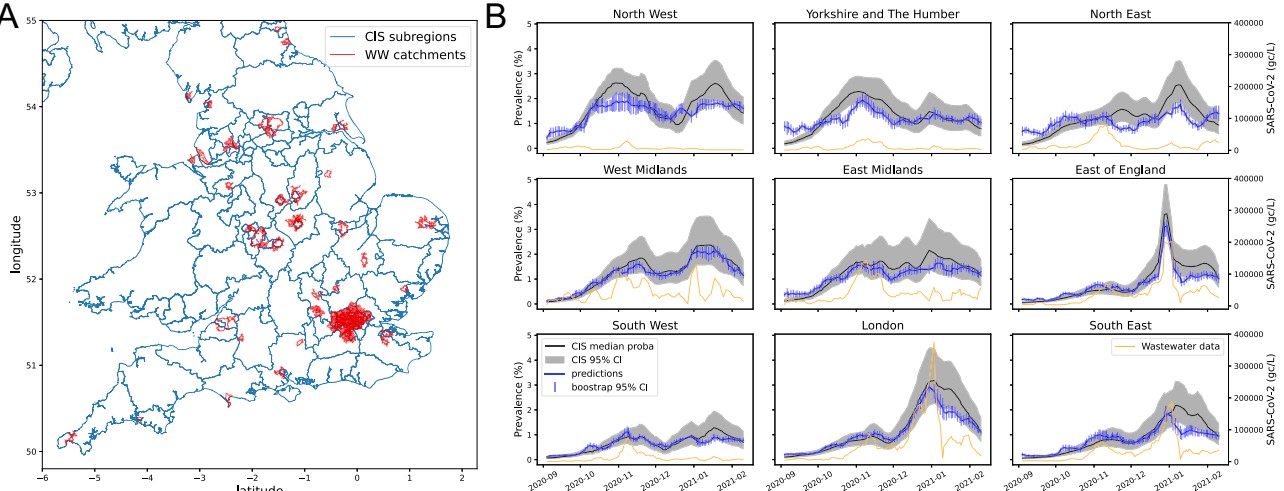

**Fig. 1 Geographical summary of the data used to estimate SARS-CoV-2 from wastewater. A** Map of Coronavirus Infection Survey (CIS) regions (outlined in blue) and the locations of wastewater (WW) catchments used in this study (in red). **B** Regional 7-day rolling averages (median) of CIS prevalence estimates (black) with 95% credible intervals using Bayesian modelling (grey regions), with corresponding predictions of prevalence from WW data only (blue) with 95% confidence interval from bootstrapping (blue vertical lines), and raw SARS-CoV-2 concentrations (yellow, right axis). The WW prevalence estimates are provided at a sub-regional level and combined to produce regional estimates for comparison.

disease dynamics and shedding where infection in a growing epidemic appears to have increased viral load[22]; inclusion of this in the model would require a non-linear model.

Combining WW data, site-level and sample-level variables within a statistical modelling framework (exemplified using a gradient boosted regression tree (GBRT) model, see the "Methods" section) to estimate prevalence of SARS-CoV-2 provides reliable metrics across regions and throughout the evolving epidemiology of the COVID-19 pandemic in England. Using this model, SARS-CoV-2 prevalence was tracked within 1.1% (with 95% confidence) from the CIS (Fig. 1B). When the GBRT model with covariates was aggregated to regional level, an average mean absolute error (MAE) was obtained, with the West Midlands performing above average and the North East performing below average (MAE of 0.12 and 0.19, respectively) (Fig. 3B). We focus the results of the modelling to a regional level here, but have carried out the analysis at a sub-regional level to inform the public health response at these smaller levels of aggregation (Fig. S6).

The GBRT model was found to be the best of the candidate models that were developed to interrogate the data and identify what variables, in addition to the raw RNA concentrations, would provide accurate estimates of prevalence. Different candidate models (linear, linear with random effects and GBRT), were evaluated and compared using the MAE between predictions and median positivity rates estimates from CIS. The addition of temporally varying data such as ammonia concentration, the fraction of samples below the limit of detection and quantification and site-specific data such as population coverage greatly improved the overall fit where GBRT the average MAE per CIS subregion reduced in value when compared with a model trained on SARS-COV-2 concentrations alone (Fig. 3A and B). Further collation of additional site-level characteristics through consultation with water companies and characterisation of the catchment area showed an additional reduction in bias in the model residuals distribution against these characteristics (Fig. S7) highlighting the robustness of our final model to wastewater network differences. While the GBRT model will be applied to estimate prevalence in England, the relative contributions of each variable and partial dependency plots (Fig. S8) are used to illustrate the direction of their effects and provide guidance for use

outside of this application. However, exploration of the site-specific random effects (within the random effects model) illustrated that there was considerable variability in MAE within sites that had yet to be fully accounted for. These WW data were collected in England across a time period where the prevalence of infection has varied considerably as a result of epidemic emergence and suppression through non-pharmaceutical interventions. The statistical modelling presented here illustrates that prevalence estimates are accurate and precise across a wide range of prevalence values (Fig. 4). The prevalence is tracked within 1.1% (with 95% confidence) for the GBRT model and is more precise at higher values of prevalence. Comparison between the random effects and GBRT model illustrates reduced precision and over estimation of prevalence at lower values of prevalence for the random effects model.

A lead and lag analysis was performed using the regression models on CIS estimates. Sampling dates for WW were shifted between −10 and 20 days with daily increments, while training a model at each step to predict the CIS positivity rates in outputs, whose dates had been fixed. At each step, the models were evaluated using the bootstrapped MAE, producing a curve of prediction errors as a function of the wastewater lag (Fig. 5). Results show a minimum value of the smoothed error curve between 0 and 2 shifted days, indicating no clear advantage to predict CIS backward or forward in time from WW data. For comparison, this analysis was replicated on Pillar 1&2 data from Test and Trace. In this case the regression outputs were the case rates reported by Test and Trace until May 17, 2021, smoothed with a 7-day centred window to remove weekly periodicity and preserve consistency of reporting dates. In addition, the WW dataset was stripped to contain samples only up to 20 days before May 17, 2021 to ensure the stability of dataset sizes during the analysis. In this case the MAE is minimal between +3 and +5 days shift, suggesting an approximate 4-day lead of WW surveillance date over reported Test and Trace cases (Fig. 5B).

## Discussion

We have shown that concentrations of SARS-CoV-2 RNA collected from wastewater in 45 sites in England, combined with essential related variables can provide reliable estimates of prevalence of SARS-CoV-2 infection within a population. Site-specific characteristics

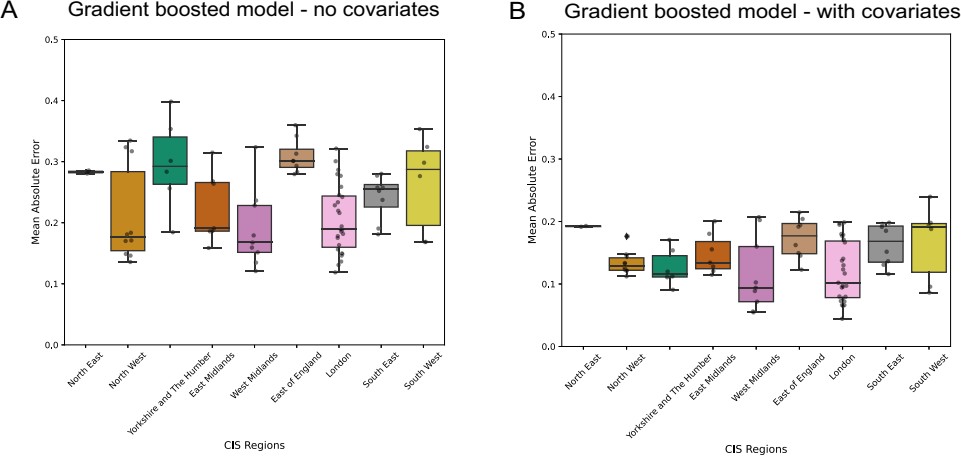

**Fig. 2 Comparison of the outputs from the phenomenological model to CIS prevalence estimates. A** Example fit between the phenomenological model estimates (green region) and the wastewater and prevalence data from three CIS sub-regions (blue dots), selected to illustrate three cases: sub-region (**A**) (good correspondence), sub-region (**B**) (concentrations tend to be low), and sub-region (**C**) (concentrations tend to be high). Model estimates of prevalence from WW data are in the same order of magnitude and follow the shape of the relationship between concentrations and prevalence using distributions of likely parameter values, but confidence intervals are wide. The combined uncertainty in parameter values exceeds the variability seen in the data. **B** The percentage of data points within each sub-region that fall within the 50% credible interval of the phenomenological model. **C** The median concentrations per positivity rate. Only CIS sub-regions that overlap with the original 44 wastewater catchment sites are shown. Sites with a poor fit to the model (yellow in sub-plot **B**) show either relatively low (dark blue) or relatively high (dark red) concentrations in sub-plot (**C**). Sites with a good fit to the model (dark green) tend to show intermediate concentrations (white).

**Fig. 3 Gradient boosted regression tree (GBRT) model performance across regions of England. A** Trained using SARS-CoV-2 concentration alone, and (**B**) including the full set of time-varying and site-specific data. Lower and upper hinges of the box plot corresponds to first and third quartile with middle line corresponding to the median. The bars indicate the 2.5th and 97.5th percentile of the values.

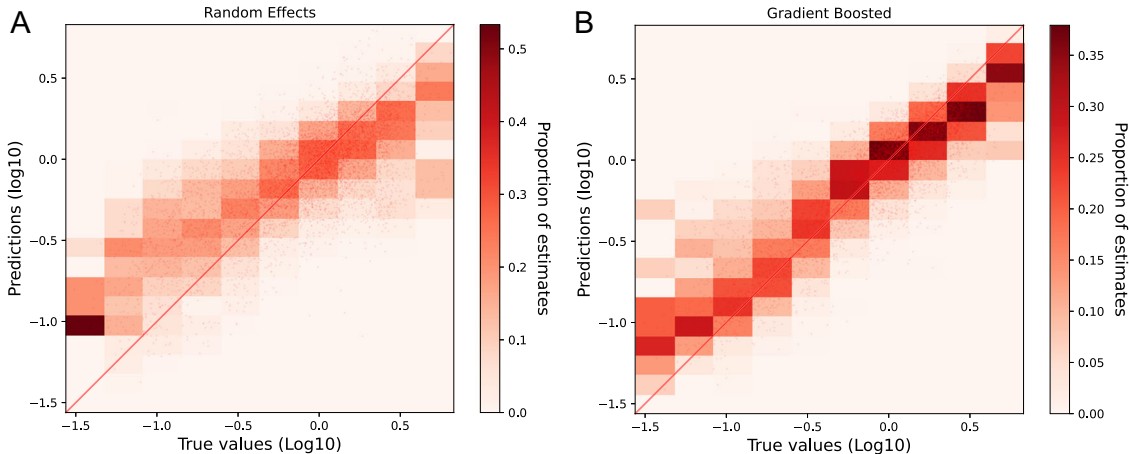

**Fig. 4 Conditional predictions of SARS-CoV-2 prevalence from WW compared to the CIS positivity estimates ("True value") in the log$_{10}$ space.** **A** Random effects model and (**B**) gradient boosted regression tree (GBRT) model. The red line indicates the diagonal where a well-fitted model would result in most predictions falling on the diagonal.

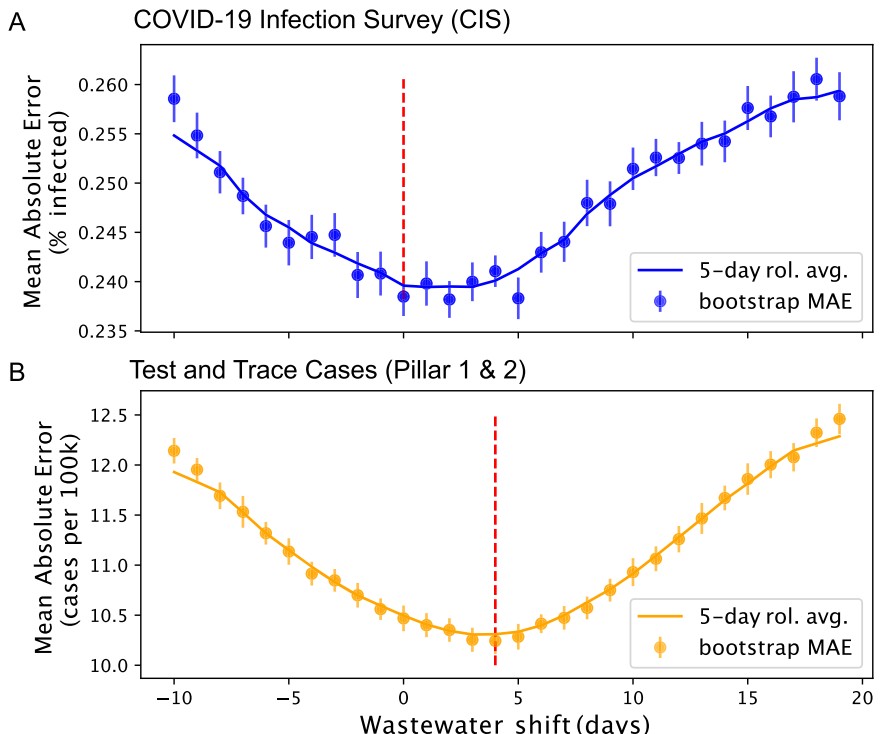

**Fig. 5 Lead and lag analysis of the WW data when compared to (A) CIS and (B) Test and Trace cases.** The shift (in days) associated with the minimal error is indicated by the red dotted line. A minimal error reached for a positive number of WW shifted days can be interpreted as a lead from wastewater by as many days. No clear lead of WW over CIS has been observed in this analysis, but an approximate 4 days lead over T&T has been observed. Vertical bars indicate the 95% confidence intervals of the mean absolute error across regions.

including limit of detection, dilution, network characteristics and other unexplained data, mean that a mechanistic model alone fails to capture the full variability in the data. However, by using locally explicit information and hierarchical models, we can understand these differences and account for them in a data-driven manner. We used our best fitting model to observe a 4-day lead in WW concentration over mainly symptomatic testing through routine surveillance. This lead in can be explained by transmission often occurring prior to symptomatic illness[23], which, it would be reasonable to assume, would be reflected in shedding of SARS-CoV-2 RNA in stool, and subsequently detected in WW. This 4-day lead illustrates the potential of WW to be an early warning tool, even in a

setting such as England in 2020–2021 which had comprehensive clinical surveillance at the time. In circumstances with limited clinical surveillance, WW can provide accurate and timely estimates of regional prevalence with just a few samples.

A strength of our analysis comes from using data on prospective surveys of infection prevalence as well as reported cases of COVID-19. The ONS CIS survey has been designed to minimise bias in prevalence estimates by incentivising participation, and accounting for under-representation of specific groups of individuals. SARS-CoV-2 is shed in faeces of both symptomatic and asymptomatic individuals and so even with perfect surveillance for cases of COVID-19 the correlation between infection and case reporting will

be imperfect due to differences in the probability of being symptomatic across regions. Furthermore, variability in reporting of symptomatic disease to the Test and Trace programme (and alternative surveillance systems specific to other countries) across socio-demographic strata is well documented[24–26], raising questions on whether clinical surveillance is an accurate and reliable estimate of disease burden. Indeed, previous comparisons of WW to disease incidence have provided variable results where under-reporting of cases has been hypothesised[7,12,27–29]. For these reasons amongst others, we argue that inferred estimates of prevalence from wastewater RNA concentrations, a measure agnostic to the source of virus, are an informative addition to clinical disease surveillance. Furthermore, WW surveillance in England has since been expanded to include additional smaller sites within networks which may enable estimates of prevalence at a smaller geographical scale than described here.

The validation of prevalence estimates derived from wastewater illustrates the critical value of collecting metadata in addition to raw RNA concentrations alone, and confirms findings from more established WW surveillance systems[30]. Several of the variables (suspended solids, ammonia concentration, phosphate) indicate organic and inorganic substances and likely approximate dilution of raw sewage with other sources of flow. The relationship is not likely to be linear as these variables also indicate the presence of agricultural runoff and so can also act as confounders. Inclusion of sample pH (with some degree of imputation in this dataset due to missingness) appears to further improve model prediction, and may reflect degradation of viral fragments at lower pH values that may reduce the sensitivity of RT-PCR, and has been previously observed in WW surveillance for poliovirus[31]. While the composite fraction, replicate samples below limit of detection/quantification, reception delay, percentage of the catchment population, the catchment area and population fraction indicate a linear relationship, their inclusion still improves model prediction. It is interesting to note that the percentage of samples from each WW site that were composite (as opposed to 'grab') had only a moderate effect on prevalence estimation. The use of composite samplers was dependent on their availability during the pandemic, with grab samples used for convenience as opposed to strategic intent. Ideally, this finding should be investigated using a comparative study design to investigate the possible added benefit of estimating SARS-CoV-2 from either approach. The probability of samples being below the limit of detection reduces as CIS positivity increases, which is to be expected, and its inclusion on the GBRT model improves the prediction of prevalence, perhaps providing further information when the estimated SARS-CoV-2 is less reliable at low prevalence. Further work will establish how these indicators should be used in settings with no measure of infection prevalence to improve inference of WW data.

This analysis has illustrated the predictive ability of WW at a time when comparatively few individuals were vaccinated against COVID-19 in England (by 1 March 2021 30% had received at least one dose). As vaccination increases the relationship between infection and faecal shedding may change and the predictive ability of the model will need to be monitored and potentially adapted to account for this. Studies of viral load in vaccinated but infected individuals have illustrated that less virus is shed in the nasopharynx by vaccinated individuals[32–34], but there are currently no data on shedding of SARS-CoV-2 viral fragments in stool from infected but vaccinated individuals. Moreover, a study of healthcare workers where the alpha variant was dominant reported no difference in viral load by vaccination status[35], and preliminary analysis from England has not identified noticeable differences in WW through to March 2021[36].

The use of WW to infer prevalence is reliant on a converging sewer network that samples a sufficient proportion of the population of interest. Remote populations, however, such as islands or rural communities, may be served by septic tank systems, leading to blind-spots in observations, especially if there is a relationship between income and centralised waste removal provision. Consequently, the benefits of wastewater-based estimates are less obvious for low density settings. Additionally, the impact of sewage effluent from hospitals has not been accounted for in this analysis which could result in an over-estimation of prevalence within a population when compared to the CIS (as hospitalised individuals are not included in the sampling). Further work will investigate the impact of hospitals and other potential sources of bias. Our analysis illustrated the added benefit of including additional metadata within a statistical model to infer prevalence highlighting the importance of site-specific characteristics. Therefore, the use of our inference model outside of the setting presented here should be avoided in the absence of further external validation and local information.

Nonetheless, close monitoring of emerging SARS-CoV-2 variants with changed phenotypic properties[37,38] will continue to be needed; meta-genomic analysis of wastewater samples provides insight into the genomic diversity of virus in the community. Also, sampling of sites that cover small catchment areas will remain capable of revealing localised spikes in incidence used to detect hotspots and inform local public health authorities. Finally, the investment that has already been made in WW surveillance systems across England, combined with insights from analysis pipelines such as ours, can be leveraged to other communicable and noncommunicable diseases and human behaviours impacting personal and community health[39,40]. As the pandemic continues to evolve, and the threat to society becomes less acute, it is likely that surveillance of SARS-CoV-2 will need to become more sustainable, making the most of those investments. These data streams will remain an important feature for public health surveillance, complementing clinical surveillance as the country emerges out of the acute phase of the COVID-19 pandemic.

## Methods

### Data

*Wastewater data from 45 sites in England.* The WW and associated metadata used for the analysis are summarised in Table S1 (a correlation matrix between variables is also provided Fig. S1). Untreated influent WW were collected from each sewage treatment works located across England. Samples were either collected as 'grab' samples (46%) or from a composite (24 h) sampler, and were transported to the laboratory and stored at 4 °C until processed (within 24 h). Physio-chemical analyses were carried out prior to further analysis. The physio-chemical analyses include quantification of pH, ammonia, orthophosphates, and suspended solids. For quantification of SARS-CoV-2 RNA from WW up to 150 ml of each sample was subjected to concentration and RNA extraction. The full details of the protocols are described in Farkas et al. [41] and Walker[42], adhere to the MIQE guidelines[43]. The use of WW as a public health tool was rapidly expanded in scope in early 2020 using the protocol in Farkas et al. where the secondary concentration step (PEG precipitation) required an 18 h incubation step. An alternative procedure was later identified with a shorter incubation step (using ammonium sulfate precipitation)[42], with equivalent results, and was adopted on the 1 January 2021. The WW quantification described by Walker includes the phage *Phi6* as a process positive control instead of PRRSV that was used in Farkas. Further details of the protocol are provided in the SI (Supplementary Note 1). Both procedures use the same extraction and RT-qPCR steps. When the impact on LoD was investigated between protocol no difference in LoD was indentified ($p = 0.356$ in an anova). The RT-qPCR assays focus on detection of the N1 and E gene, and here our analyses is on quantification of the N1 gene (see the SI for details of the primer used). A 10-fold dilution series of RNA standards within the range of $10^4$–$10^1$ gene copies(gc)/μL was included on each RT-qPCR plate to generate a standard curve. Standard curves were accepted if the slope of the $\log_{10}$ RNA standard concentration versus $C_q$ was between $-3.1$ and $-3.6$ and if the $r^2$ for the curve was >0.98, a summary table of these data are presented in the SI. For each sample two replicate $C_q$ values were used to calculate the gc/l in the original sample, based on the standard curves.

The limit of detection (LoD—the lowest concentration where all replicates were positive) and limit of quantification (LoQ—the lowest concentration where the coefficient of variance was below 0.25) were determined by running WW extracts (devoid of RNA) spiked with nominal concentrations of SARS-CoV-2 ranging from 100 to 2 gc/μL in replicates of 10. For the N1 gene the LoD was 1.7 gc/μL and the LoQ was 11.8 gc/μL in the protocol described by Farkas[41], and the protocol

described by Walker the LoD was 0.4 gc/μL and the LoQ was 4 gc/μL. Note that these estimates of LoD and LoQ should be regarded as theoretical limits, and in practice the limits are likely to be higher and vary. Figure S5 illustrates that as CIS positivity increases replicate samples below LoD are less likely, and there is some evidence of site-specific variability, which is the subject of further investigation.

Of the total 6228 samples supplied from the lab in the time-frame of this analysis, 1365 samples returned a value of 'NA' for SARS-CoV-2 (meaning that the submitted sample did not provide meaningful results for further use), and were removed from the analysis, leaving 4863 observations that were taken forward. Of these observations, 24.5% of replicate samples were below the LoD and 33.7% were below the LoQ; these values were retained in the analysis. Finally, a $log_{10}$ transformation was applied to all the concentration variables and the target variables to reduce the heavy skewness of the distribution.

*The ONS Coronavirus infection survey.* The ONS COVID-19 infection survey data are used to infer subregional estimates of positivity[3]. CIS is a large household survey with longitudinal follow-up (ISRCTN21086382). The study received ethical approval from the South Central Berkshire B Research Ethics Committee (20/SC/0195). Private households are randomly selected on a continuous basis from address lists and previous surveys to provide a representative sample across the UK. For the current study, only data from England was used. At the first visit, participants were asked for (optional) consent for follow-up visits every week for the next month, then monthly for 12 months from enrolment. At each visit, enroled household members provided a nose and throat self-swab following instructions from the study worker. The CIS was designed to test 150,000 people every 2 weeks across England in October 2020, and this sample size was designed to correspond with 15,000–20,000 individuals in each of the nine governmental office regions (North East, North West, Yorkshire and the Humber, East Midlands, West Midlands, East of England, London, South East, South West), providing an approximate 0.1%, 0.2%, and 0.5% margin of error on 0.1%, 0.5%, and 2%, respectively. In September 2020 the study design was adapted to have sufficient power to estimate prevalence at a sub-regional level, resulting in further increase in sample size of approximately 4-fold.

For the time periods relevant to this study (July 2020–March 2021), the number of participants per two-week period varied from 31,294 to 183,167 with an average of 126,655 participants, and typically up to 90% of participants had at least 5 visits. These participants were recruited from approximately 64,586 (range 14,965–93,940) households within any 2-week period. For Further details are provided in the statistical bulletins provided by the Office for National Statistics (https://www.ons.gov.uk/peoplepopulationandcommunity/healthandsocialcare/conditionsanddiseases/bulletins/coronaviruscovid19infectionsurveypilot/previousReleases).

*Linkage of WW data to CIS data.* Individual level data from the CIS are not available due to confidentiality agreements, and so subregional estimates of positivity are the most geographically precise estimates of SARS-CoV-2 prevalence available. Within England the nine regions are divided into 119 sub-regions. The subregional estimates of positivity were computed weekly, and made publicly available (For example, see https://www.ons.gov.uk/peoplepopulationandcommunity/healthandsocialcare/conditionsanddiseases/bulletins/coronaviruscovid19infectionsurveypilot/19november2021#sub-national-analysis-of-the-number-of-people-who-had-covid-19 "Sub-regional analysis for the UK").

To link the CIS sub-regions to wastewater data, we need to make the statistical assumption that wastewater data are uniformly distributed within sub-regions. Thus, prevalence estimates from wastewater should be an unbiased estimate from the general population within each subregion. In order to implement this the wastewater catchment areas were mapped onto CIS sub-regions. Typically, the catchment area for each treatment plant is smaller in size than a CIS sub-region, so the catchment collects wastewater from just one sub-region (this is the case for 56 of the 83 sub-regions), but there can be many catchments per sub-region (a schematic example is given in Fig. S3, see the example of sub-region C and the numerous catchments within this region). However, for some regions of England (especially in the greater London area) catchment areas cover multiple sub-regions. The mapping of the catchment areas to geographical areas were made available by the water companies. The result of the mapping is that each site-level dataset has a corresponding CIS sub-region assigned to it. Where sites covered multiple sub-regions, the data were duplicated and each duplicate assigned a sub-region with a corresponding portion of the sub-region covered. These proportions are used later when estimating prevalence at a sub-region (and region) level, where the calculated proportions are used as weights.

A total of 4863 wastewater samples are available for the 45 sites within the dataset (over 214 days), corresponding to approximately 3 samples per site per week. These values were linearly interpolated to daily estimates for each site. These estimates were then merged onto the CIS dataset with an average gc/l for each daily estimate of CIS positivity, where multiple sites were combined using the calculated weights. With the details of the number of unique properties included or not in each LSOA/catchment intersection, the subregional coverage (population covered per WW site) is then inferred.

*The Test & Trace Pillar 1&2 case rates at Layer Super Output Areas (LSOA) level.* Case rates of the number of new people infected per 100,000 individuals from Pillar 1&2 data were available form Public Health England. These data were aggregated

to wastewater catchment level using a mapping from LSOA to catchment areas. Furthermore, the case rates were smoothed using a 7-day moving average to remove any artefactual weekly periodicity and therefore provide a better estimate of incidence. The same mapping procedure described for the CIS data is used here.

## Models

*Phenomenological model.* The phenomenological model considers that prevalence (*P*, or the proportion of population infected) is related to *C* (the measured viral concentration in the sample, in units of gene copies per litre), by (Eq. (1)):

$$P = \frac{C \times Q_P}{S \times V},  \quad (1)$$

where $Q_p$ is the wastewater generated by one person per day (in L/day), *S* is the mean shedding rate, or the concentration of virus in stool of infected people (in gc/ml) and *V* is the mean volume of stool per person per day (in ml/day). A similar equation is used in ref. [19] using total flow rather than per person flow. We use the same model for all sites. Mean flow is set to 400 L/person/day, based data from 15 sites where flow is routinely collected. Mean shedding rate is assumed to be $1.9 \times 10^6$ gc/mL from ref. [20]. A mean stool volume of 128 g/person/day is assumed based on a review of 95 clinical studies—the majority being UK-based[44], and factor of 1.06 ml/g is used to convert from g to ml[45].

The model is used to identify which variables result in a considerable variability in the output (*P*). Two methods were utilised in the sensitivity analysis; variance based sensitivity analysis (VBSA) and PAWN. The VBSA is a global sensitivity analysis where the variance of the output is decomposed into fractions attributed to the inputs. The PAWN approach considers the entire distribution of the outcome (using the cumulative distribution function), which can be useful in cases where variance is not an adequate proxy of uncertainty.

*Spatiotemporal analysis to obtain CIS prevalence estimates.* Bayesian multilevel regression and poststratification (MRP) is an increasingly used statistical technique to obtain representative estimates of prevalence or preferences at the national and smaller regional levels[3]. By using random effects in the multilevel model stable estimates can be obtained for subnational levels from relative small samples or relatively rare outcomes. However, if there is an underlying spatial structure this needs to be captured by the MRP methodology to avoid biased estimates based on a model that assumes independent group-level errors. Gao et al.[46] recently proposed a spatial MRP using a Besag–York–Mollié specification for the regional effect.

Here we extended the spatial MRP approach proposed by Gao et al. to a spatio-temporal context by adding a temporal component to the model. For the temporal components we use autoregressive or random walk processes with discrete time indices (weeks) to capture likely temporal effects in the MRP model. The choice of the type of directed conditional distribution for the time effect (random walk or autoregressive) type of space–time interaction (type I–IV[47]), and inclusion of additional covariates was guided by comparing the Watanabe-Akaike information criterion (WAIC) of the models. A type I space–time interaction, which assumes no spatial and/or temporal structure on the interaction, with first-order autoregressive terms were selected based on the WAIC.

The following covariates and interactions were considered for the MRP: age (2–11, 12–16, 17–24, 25–34, 35–49, 50–69, 70+); sex; ethnicity (white/non-white), CIS area; region (9 regions in England); time and two-way interactions of age and time, ethnicity and time, area and time, region and ethnicity, and region and age. After running the spatiotemporal regression model, post-stratification was used to obtain representative estimates of the outcome prevalence in the target population. Post-stratification tables were based on the conditional distribution of age and sex by area from ONS. The conditional distribution of ethnicity by these categories were obtained from the ETHPOP database[48]. Using the population sizes of each poststratification cell of the target population, MRP adjusts for residual non-representative by post-stratifying by the percentage of each type in the actual overall population[46]. The outputs of these analyses consist of median estimates of percent positivity rates and associated 95% credible intervals available weekly at CIS subregional aggregation level between August 31, 2020 and February 14, 2021. For application in this study the estimates were up-sampled daily by linear interpolation between weekly estimates for each subregion and joined to the wastewater dataset using mappings from Lower Tier Local Authority (LTLA) to CIS subregions and from LTLA to catchment areas. Note that the resulting joined dataset was at CIS subregional level, where STW's contributions were weighted by their population covered in each overlapping subregion.

*Modelling the population prevalence from WW and associated metadata.* The prevalence of SARS-CoV-2 at a sub-regional level was estimated using WW and associated metadata. A set of candidate statistical models were used to examine the relationship of RNA concentration to the median of the posterior estimate of the CIS positivity rate. These were: (1) linear regression, (2) linear regression with random effects intercept, (3) linear regression with random effects intercept and slope, (4) gradient boosted regression tree (machine learning) models. Their out-of-sample predictive ability used to determine which model most effectively estimates prevalence.

Random effects models: This model is a linear model where WW and the metadata have random effects on the slope and the intercept. The Python *statsmodels* package was used to implement these models. For a Bayesian description of the model with another application case in WBE (see ref. [36]). The linear regression and linear regression with random effects intercept had the poorest performance and are not described further.

Gradient Boosted Regression Trees: This model consists of a linear combination of non-linear predictors (also known as decision trees) trained by gradient descent[49]. Its performance has been shown on many regression examples[50–52], and it is especially good at combining a large number or variety of input variables in a non-linear way. The implementation chosen here is an "Extreme Gradient Boosting" from python *XGBoost* package, which simply refers to an efficiently optimised Gradient Boosting Trees regressor using second-order optimisers.

*Model evaluation.* The models were compared using the mean absolute error (MAE) from out-of-sample prediction (to limit over-fitting of the models). The out-of-sample prediction was carried out using repeated random sampling: 50 random splits were generated in the available dataset, in each sample 80% of the data were retained for training and the remaining 20% for testing. The MAE was generated for each dataset and the combined MAE were obtained by averaging the test results from all samples. If $y_i, (i = 1, \ldots , n)$ are the testing data for sample $i$, and $\hat{y}_i$ the predictions from the model, the mean absolute error associated with the $i$th test set of size $n$ is

$$\mathrm{MAE}_i = \frac{1}{n} \sum_{i=1}^{n} |10^{y_i} - 10^{\hat{y}_i}| \qquad (2)$$

The estimated MAE, and associated standard deviation and 95% confidence intervals are then computed as

$$\mathrm{MAE} = \frac{1}{N_{\mathrm{samples}}} \sum_{i \in \mathrm{samples}} \mathrm{MAE}_i \qquad (3)$$

$$\sigma_{\mathrm{MAE}} = \sqrt{\frac{\sum_{i \in \mathrm{samples}} \left( \mathrm{MAE}_i - \mathrm{MAE} \right)^2}{N_{\mathrm{samples}}}} \qquad (4)$$

$$\mathrm{CI}_{95\%} \approx \left[ \frac{-2\sigma_{\mathrm{MAE}}}{\sqrt{N_{\mathrm{samples}}}}, \frac{2\sigma_{\mathrm{MAE}}}{\sqrt{N_{\mathrm{samples}}}} \right] \qquad (5)$$

*Model performance across STWs' characteristics.* In October and November 2020, interviews were conducted with nine water utilities in England to document information on sewer network characteristics that could impact model performance (average daily flow, proportion of pumping in catchment, combined vs foul sewers etc.) The knowledge gathered from the interviews and a related questionnaire provided both qualitative and quantitative data, and transformed into variables used to assess model performance. An average MAE was obtained for each wastewater treatment plant by averaging the scores of CIS subregions included in the catchment weighted by the population covered. These resulting WTP MAE were then plotted across sites characteristics (Fig. S7).

**Reporting summary**. Further information on research design is available in the Nature Research Reporting Summary linked to this article.

## Data availability
The data used in this study are available online https://www.gov.uk/government/publications/monitoring-of-sars-cov-2-rna-in-england-wastewater-monthly-statistics-15-july-2020-to-30-march-2022

The full data that support the findings of this study are available alongside the code, within the repository provided in the code availability statement.

## Code availability
The specific data and code used in this study is available at https://github.com/kath-o-reilly/wbe_prevalence_england_python and https://zenodo.org/badge/latestdoi/476033528.

This study made use of Numpy, Pandas, Sklearn, Statsmodels, and XGBoost python open packages. Python (v 3.7.4) was implemented using Jupyter notebooks (v 6.0.1). Some plotting and summary statistics were implemented using R (v 4.1.2) and Rstudio (v 2022.02.3).

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

## Acknowledgements

We are grateful to Professor Rowland Kao, and Dr. Chris Jewell for helpful discussions in model development and interpretation, and for members of the "SPI-M" COVID-19 group for helpful feedback as the modelling developed. We are extremely grateful to representatives from the many water companies in England that provided insight into their sewage treatment works, and provided catchment area data for use in the analysis. This manuscript benefitted greatly from input provided during peer review. This study was predominantly funded by UKHSA, and KP acknowledges funding from UK Natural Environment Research Council National under the COVID-19 Wastewater Epidemiology Surveillance Programme (NE/V010441/1). The mapping within Fig. 2 was facilitated through OpenStreetMap, under the Open Database License (CC BY-SA) (https://www.openstreetmap.org/copyright).

## Author contributions

M.M., A.L.J, C.S., K.P., M.W., J.G., K.M.O. and L.D. conceptualised the analysis and were involved in the development of the approach. M.M., A.J.L., C.S., C.L. and K.M.O. analysed the data and generated the results. All authors (M.M., A.L.J., C.S., M.W., T.H., K.P., C.L., A.S., J.P., N.E., D.I.W., J.T.B., A.E., J.G., K.M.O., L.D.) contributed to writing the first and subsequent versions of the manuscript and have approved the manuscript for peer-review. D.W. provided guidance on the use of the laboratory methods and their interpretation. J.P. and N.E. developed and implemented the laboratory protocols and work. The views expressed in this paper are those of the authors and do not necessarily reflect the views or policies of the Department of Health and Social Care or Agencies of the authors.

## Competing interests

The authors declare no competing interests.
