## [Peer Review File · Nature Communications]

An analysis of 45 large-scale wastewater sites in England to estimate SARS-CoV-2 community prevalenceREVIEWER COMMENTS

Reviewer #1 (Remarks to the Author):

The authors have compared levels of SARS-CoV-2 quantified in wastewater to prevalence estimates from continuous surveys in England. It is an important question and timely, and the study appears to have access to data necessary to address the question. However, it appears that the authors' conclusions are built upon a great deal of data imputation, both across space and over time. Given the short-lived nature of a COVID-19 infection (< 14 days) as well as the geospatial heterogeneity, I am quite concerned that this data imputation is adversely affecting the analysis and generating too much error around the results.

Specific comments:

- 1) Lines 46-47 the authors suggest 22-98% of COVID-19 cases result in symptomatic disease, with an average of 75%. This is a gross overestimate. The Alene article cited estimated 28-31%. Sah's recent systematic review suggests about 37%. I recommend using these estimates and references rather than the Subramanian article which is biased in its estimate due to convenience sampling.
- Sah P, Fitzpatrick MC, Zimmer CF, Abdollahi E, Juden-Kelly L, Moghadas SM, et al. Asymptomatic SARS-CoV-2 infection: A systematic review and meta-analysis. *Proc Natl Acad Sci USA*. 2021 Aug 24;118(34):e2109229118.
- 2) Line 56-57. Medema et al. were the first to show proof of concept and should be cited here.
- Medema G, Heijnen L, Elsinga G, Italiaander R, Brouwer A. Presence of SARS-Coronavirus-2 RNA in Sewage and Correlation with Reported COVID-19 Prevalence in the Early Stage of the Epidemic in The Netherlands. *Environ Sci Technol Lett*. 2020 Jul 14;7(7):511-6.
- 3) Line 520. Can the authors specify what the further secondary concentration was? It looks like they were centrifuged, but at what G and for how long?
- 4) Line 524. Should be RNA rather than DNA?
- 5) Lines 527-530. The authors state that they spiked wastewater samples with low concentrations of SARS-CoV-2. How does this help the authors obtain an LOD and LOQ -there may or may not have been SARS-CoV-2 in the sample before they spiked it?
- 6) Line 531-532. The authors here appeared to replace samples without detection of SARS-CoV-2 with mean value over the "week number and sites". Did the authors just impute the mean value for the entire study area? This is quite problematic - why not simply use a non-detection when SARS-CoV-2 was not detected?
- 7) Lines 532-535. I'm quite concerned with the data imputation. Getting a result below the limit of detection is still a result, and should not be changed. See for example: Larsen DA, Collins MB, Du Q, Hill D, Insaf TZ, Kilaru P, et al. Coupling freedom from disease principles and early warning from wastewater surveillance to improve health security. *medRxiv*. 2021 Jun 14;2021.06.11.21258797.
- 8) Lines 537-546. The authors need to better describe the ONS Coronavirus Infection Survey. What was the sample size? How was it powered? How many households per week and month?
- 9) How have the authors geographically matched the survey data to the sewershed data? Lines 548-556 suggest that the authors simply allowed the subregion to match the sewershed if they were somehow overlapping. Do the authors not have access to the geocoordinates of the CIS participants, and can they not then match those to the sewershed boundaries?
- 10) Lines 620-622. The authors weight the catchments within the sewersheds by the size of the catchment if there are more than one catchment within a sewershed. Again, why don't the authors simply use the geocoordinates of the randomly sampled houses to get more precise estimates of prevalence within the sewershed rather than trying to mash together the catchments with the sewersheds?
- 11) How is prevalence defined in the CIS? The authors should include that.
- 12) Lines 641-644. I am confused at the out-of-sample prediction and the bootstrapping here. The authors should simply split their data into training and testing. Why have the authors done 50 random splits and bootstrapped 80% of the data? It's confusing and raises concerns that the authors have duplicated their data numerous times to obtain statistical power. I would love to see is a simple figure showing from the testing data the modeled estimates of prevalence and measured estimates of prevalence.

Reviewer #2 (Remarks to the Author):

Morvan et al. present a manuscript detailing wastewater monitoring for SARS-CoV-2 at 44 sites in England. This effort is notable for its large size relative and population coverage. Parameters including water quality metrics and flow rates are monitored as well. The authors then apply the developed data to a model to match prevalence estimates with independently determined clinical disease prevalence.

Overall, the work is technically well completed. The primary contribution of this study is its breadth and modeling insights.

The modeling approach appears to be driven to maximize the accuracy of results compared to prevalence surveys, which appears to be a fitting exercise rather than yielding any specific fundamental or mechanistic insights. The presentation of the modeling approach and results is not clear, so this may be addressed by revision for presentation. It is also unclear if the modeling results would be extensible to other sites or systems, or what considerations would be for extending the model further.

The paper also dives into the potential for using this system as an 'early warning' tool; however, there has been significant debate about this in the literature recently (some citations below). I would suggest revising these claims to include all appropriate considerations/caveats, and perhaps removing them from the abstraction

Olesen, Scott W., Maxim Imakaev, and Claire Duvallet. "Making Waves: Defining the lead time of wastewater-based epidemiology for COVID-19." *Water Research* (2021): 117433.

Bibby, Kyle, Aaron Bivins, Zhenyu Wu, and Devin North. "Making waves: Plausible lead time for wastewater based epidemiology as an early warning system for COVID-19." *Water Research* (2021): 117438.

Authors should include all relevant qPCR details (MIQE guidelines).

The validity of replacing non-detect values with weekly means is unclear. This assumes that non-detects were in error rather than a true result due to variation in the signal. This needs to be better justified or an alternative approach taken.

The authors cluster grab and composite samples but do not make a clear attempt to unify them. At a minimum I suggest discussion of the variability that this approach might take, or potentially an analysis of sample variability between these groups.

Response to reviewers comments for “**Estimating SARS-CoV-2 prevalence from large-scale wastewater surveillance: insights from combined analysis of 44 sites in England**” (NCOMMS-21-30150)

Editorial requirements:

- Editorial policy checklist: attached in revision
- Reporting summary: attached in revision
- Data Availability: further detail provided in manuscript
- Code Availability: provided in manuscript, see https://github.com/kath-o-reilly/wbe_prevalence_england_python
- ORCID: details for all authors have been encouraged

REVIEWER COMMENTS

Reviewer #1 (Remarks to the Author):

The authors have compared levels of SARS-CoV-2 quantified in wastewater to prevalence estimates from continuous surveys in England. It is an important question and timely, and the study appears to have access to data necessary to address the question. However, it appears that the authors’ conclusions are built upon a great deal of data imputation, both across space and over time. Given the short-lived nature of a COVID-19 infection (< 14 days) as well as the geospatial heterogeneity, I am quite concerned that this data imputation is adversely affecting the analysis and generating too much error around the results.

We thank the reviewer for the positive assessment of the importance of our work and thoughtful comments which we address below. This additional work has substantially improved the revised version as we provide additional evidence to support our results in a more robust way.

We think that the reviewers concerns about imputation may have been due to poor communication on our part of what has been imputed, for example by not giving concrete numbers of imputed values. These numbers are low, and for clarification we provide these numbers where relevant. We hope that the point-by-point comments addresses the reviewers concerns.

Specific comments:

1) Lines 46-47 the authors suggest 22-98% of COVID-19 cases result in symptomatic disease, with an average of 75%. This is a gross overestimate. The Alene article cited estimated 28-31%. Sah’s recent systematic review suggests about 37%. I recommend using these estimates and references rather than the Subramanian article which is biased in its estimate due to convenience sampling.

- Sah P, Fitzpatrick MC, Zimmer CF, Abdollahi E, Juden-Kelly L, Moghadas SM, et al.

Asymptomatic SARS-CoV-2 infection: A systematic review and meta-analysis. Proc Natl Acad Sci USA. 2021 Aug 24;118(34):e2109229118.

Apologies, we think the distinction between symptomatic (referred to in the text) and asymptomatic (referenced in the publications) may have been confused. We are happy to exclude the reference to Subramanian et al. and Alene et al. and include the additional meta-analysis from Sah et al., as the potential for bias is an important one. [ref given page 3, line 8, ref 8]

2) Line 56-57. Medema et al. were the first to show proof of concept and should be cited here.

- Medema G, Heijnen L, Elsinga G, Italiaander R, Brouwer A. Presence of SARS-Coronavirus-2 RNA in Sewage and Correlation with Reported COVID-19 Prevalence in the Early Stage of the Epidemic in The Netherlands. Environ Sci Technol Lett. 2020 Jul 14;7(7):511–6.

Thank you for the suggestion, we have replaced the Ahmed citation with Medema et al. (2020), ref 13.

3) Line 520. Can the authors specify what the further secondary concentration was? It looks like they were centrifuged, but at what G and for how long?

We apologise for omitting this information. Since submission a full protocol of the methods used have been published by Farkas et al, and an amended protocol has since been developed by Walker et al. As the analysis makes use of data from both of these protocols we have been more explicit about the differences in protocol and when the methods by Walker was implemented (01 Jan 2021).

Below is the amended text in the Methods section:

[page 17, line 11] "For quantification of SARS-CoV-2 RNA from WW up to 200 ml of each sample was subjected to concentration and RNA extraction. The full details of the protocols are described in Farkas et al. and Walker et al., and adhere to the MIQE guidelines (REF). The use of WW as a public health tool was rapidly expanded in scope in early 2020 using the protocol in Farkas et al. where the secondary concentration required an 18 hour incubation step (PEG precipitation). An alternative procedure was later identified (using ammonium sulphate precipitation, Walker et al 2021) that required a much shorter incubation step, with equivalent results, and was adopted on the 1st January 2021. Additionally, the WW quantification described by Walker includes the phage Phi6 as a process positive control instead of PRRSV that was used in Farkas. Further details of the protocol are. Provided in SI, along with data illustrating the equivalence of the concentration steps. Both procedures use the same RT-qPCR step."

Please see the amended material in the SI for further details of the two protocols. Here we describe the centrifugation; in Farkas samples were centrifuged at 3000 x g for 30 mins, and in Walker samples were centrifuged at 10,000 x g for 30 mins.

4) Line 524. Should be RNA rather than DNA?

Thanks for spotting this typo - fixed.

5) Lines 527-530. The authors state that they spiked wastewater samples with low concentrations of SARS-CoV-2. How does this help the authors obtain an LOD and LOQ – there may or may not have been SARS-CoV-2 in the sample before they spiked it?

We apologise that our description was not clear. We have provided further details in the track-changed document (included below). The WW samples used in the LoD and LoQ tests were extracts from the supernatant in the first concentration step, and consequently would not contain virus.

[page 17, line 24] “The limit of detection (LoD - the lowest concentration where all replicates were positive) and limit of quantification (LoQ - the lowest concentration where the coefficient of variance was below 0.25) were determined by running WW extracts (devoid of RNA) samples spiked with nominal concentrations of SARS-CoV-2 ranging from 100 to 2 gc/μL in replicates of ten. For the N1 gene the LoD was 1.7 gc/μLgc L-1 and the LoQ was 11.8 gc/μL in the protocol described by Farkas, and the protocol described by Walker the LoD was 0.4 gc/μL and the LoQ was 4 gc/μL.”

6) Line 531-532. The authors here appeared to replace samples without detection of SARS-CoV-2 with mean value over the “week number and sites”. Did the authors just impute the mean value for the entire study area? This is quite problematic – why not simply use a non-detection when SARS-CoV-2 was not detected?

We have added further details of methods for the analysis of the data. To avoid confusion, we do not impute values below LOD/LOQ from ‘nearby’ data. The description of the raw data now in the methods (and summary table S1) should clarify this question,

[page 18, line 8] “Of the total 6,228 samples supplied from the lab in the time-frame of this analysis, 1,365 samples returned a value of ‘NA’ for SARS-CoV-2, and were removed from the analysis, leaving 4,863 observations that were taken forward. Of these observations, 24.5% of replicate samples were below the limit of detection and 33.7% were below the limit of quantification; these values were retained in the analysis.”

7) Lines 532-535. I’m quite concerned with the data imputation. Getting a result below the limit of detection is still a result, and should not be changed. See for example: Larsen DA, Collins MB, Du Q, Hill D, Insaf TZ, Kilaru P, et al. Coupling freedom from disease principles and early warning from wastewater surveillance to improve health security. medRxiv. 2021 Jun 14;2021.06.11.21258797.

We hope that the response to the previous point addresses this concern.

The reviewers comments regarding LOD/LOQ, and internal discussions have prompted some further investigations to determine the ‘field LOD’ by comparing the CIS prevalence to the proportion of replicate samples that are below LOD and comparing the finding by site, see fig. S5. We have found (as expected) an increasing % of replicate above LOD for increasing CIS prevalence and some evidence of site specific effects, and this field LOD is above the lab-determined LOD. We agree that this is an important aspect of ww to investigate, but falls

outside of the scope of this current study, and are currently investigating this further. We would be happy to correspond directly with the reviewer on this, if interested.

8) Lines 537-546. The authors need to better describe the ONS Coronavirus Infection Survey. What was the sample size? How was it powered? How many households per week and month?

We have provided further information as requested;

[page 18, line 22] “The CIS was designed to test 150,000 people every two weeks across England in October 2020, and this sample size was designed to correspond with 15,000-20,000 individuals in each of the nine governmental office regions (North East, North West, Yorkshire and the Humber, East Midlands, West Midlands, East of England, London, South East, South West), providing an approximate 0.1%, 0.2% and 0.5% margin of error on 0.1%, 0.5% and 2%, respectively. In September 2020 the study design was adapted to have sufficient power to estimate prevalence at a sub-regional level, resulting in further increase in sample size of approximately 4-fold.

For the time periods relevant to this study (July 2020 – March 2021), the number of participants per two-week period varied from 31,294 to 183,167 with an average of 126,655 participants, and typically up to 90% of participants had at least 5 visits. These participants were recruited from approximately 64,586 (range 14,965 to 93,940) households within any two-week period. Further details are provided in the statistical bulletins provided by the Office for National Statistics (<https://www.ons.gov.uk/peoplepopulationandcommunity/healthandsocialcare/conditionsanddiseases/bulletins/coronaviruscovid19infectionsurveyypilot/previousReleases>).

9) How have the authors geographically matched the survey data to the sewershed data? Lines 548-556 suggest that the authors simply allowed the subregion to match the sewershed if they were somehow overlapping. Do the authors not have access to the geocoordinates of the CIS participants, and can they not then match those to the sewershed boundaries?

We have provided further clarification of the linkage of WW data to CIS data. Confidentiality of CIS participants prevents household mapping so we needed to take the alternative approach of modelling at a sub-regional level and apply weighting to the WW data. We have provided further details in the Methods and a supplementary figure to help explain this.

“[page 19, line 12] Individual level data from the CIS are not available due to confidentiality agreements, and so subregional estimates of positivity are the most geographically precise estimates of SARS-CoV-2 prevalence available. Within England the nine regions are divided into 119 sub-regions. The subregional estimates of positivity were computed weekly, and made publicly available (For example see <https://www.ons.gov.uk/peoplepopulationandcommunity/healthandsocialcare/conditionsanddiseases/bulletins/coronaviruscovid19infectionsurveyypilot/19november2021#sub-national-analysis-of-the-number-of-people-who-had-covid-19> “Sub-regional analysis for the UK”).

To link the CIS sub-regions to wastewater data, we need to make the statistical assumption that wastewater data are a random sample from individuals within each sub-region. Thus, prevalence estimates from wastewater should be an unbiased estimate from the general population within each subregion. In order to implement this the wastewater catchment areas were mapped onto CIS sub-regions. Typically, the catchment area for each treatment plant is smaller in size than a CIS sub-region, so the catchment collects wastewater from just one sub-region (this is the case for 56 of the 83 sub-regions), and there can be many catchments per sub-region (a schematic example is given in Fig. S3, see the example of sub-region C and the numerous catchments within this region). However, for some regions of England (especially in the greater London area) catchment areas cover multiple sub-regions. The mapping of the catchment areas to geographical areas were made available by the water companies, and the mapping carried out. To account for this the specific sites were mapped onto CIS sub-regions, resulting in each site-level dataset having a corresponding CIS sub-region assigned to it. Where sites covered multiple sub-regions, the data were duplicated and each duplicate assigned a sub-region with a corresponding portion of the sub-region covered. These proportions are used later when estimating prevalence at a sub-region (and region) level, where the calculated proportions are used as weights.”

10) Lines 620-622. The authors weight the catchments within the sewersheds by the size of the catchment if there are more than one catchment within a sewershed. Again, why don't the authors simply use the geocoordinates of the randomly sampled houses to get more precise estimates of prevalence within the sewershed rather than trying to mash together the catchments with the sewersheds?

We hope that the response to the earlier question address this question.

11) How is prevalence defined in the CIS? The authors should include that.

We now provide an explicit definition of positivity in the Introduction;

[page 4, line 23] “The Covid Infection Survey (CIS) was established across England and Wales by the Office of National Statistics early in the pandemic to assess the prevalence of individuals in the community testing positive for SARS-CoV-2 (otherwise known as “positivity”) through stratified nasopharyngeal sampling of individuals in households”

12) Lines 641-644. I am confused at the out-of-sample prediction and the bootstrapping here. The authors should simply split their data into training and testing. Why have the authors done 50 random splits and bootstrapped 80% of the data? It's confusing and raises concerns that the authors have duplicated their data numerous times to obtain statistical power. I would love to see is a simple figure showing from the testing data the modeled estimates of prevalence and measured estimates of prevalence.

We apologise that the approach was not communicated well. To ensure a realistic predictive ability we split our train (80%) and test (20%) data and carry out out-of-sample predictions on the test data. If we were to repeat this using standard approaches, we would only be able to perform a ‘5-fold’ out-of-sample prediction. To minimise bias of the procedure we instead opted for a random 80% of samples and repeated this 50 times, hence using the bootstrap

terminology. We believe that the estimates provide a more realistic estimate of out-of-sample performance. To avoid confusion we have omitted the term 'bootstrap'

[page 25, line 3] "The out-of-sample prediction was carried out using repeated random sampling: 50 random splits were generated in the available dataset, in each sample 80% of the data were retained for training and the remaining 20% for testing. The MAE was generated for each dataset and the combined MAE were obtained by averaging the test results from all samples. If $y_i, (i=1, \dots, n)$ are the testing data for sample i , and (\hat{y}_i) the predictions from the model, the mean absolute error associated with the i^{th} test set of size n is:"

Reviewer #2 (Remarks to the Author):

Morvan et al. present a manuscript detailing wastewater monitoring for SARS-CoV-2 at 44 sites in England. This effort is notable for its large size relative and population coverage. Parameters including water quality metrics and flow rates are monitored as well. The authors then apply the developed data to a model to match prevalence estimates with independently determined clinical disease prevalence.

Overall, the work is technically well completed. The primary contribution of this study is its breadth and modeling insights.

We thank the reviewer their insightful comments, and enthusiasm for the research undertaken.

1) The modeling approach appears to be driven to maximize the accuracy of results compared to prevalence surveys, which appears to be a fitting exercise rather than yielding any specific fundamental or mechanistic insights. The presentation of the modeling approach and results is not clear, so this may be addressed by revision for presentation. It is also unclear if the modeling results would be extensible to other sites or systems, or what considerations would be for extending the model further.

We have worked considerably on the text to ensure that the description of the data and analysis is clear. We hope the concerns of the reviewer have been addressed. The motivation behind the study is to illustrate the utility of wastewater data by validating the approach against the current best practice; population surveys carried out by ONS. We have learnt that with appropriate analysis of data the prevalence of SARS-CoV-2 in the population can be accurately estimated from wastewater. Further, inclusion of meta-data along with raw estimates of gc/l further improves estimates.

We agree with the reviewer that application of the approach may be challenging outside of the current area, and within the dataset we use out of sample prediction to examine this. As part of further research being undertaken in 2021-22 we have seen that WW requires adaption to estimate prevalence, possibly due to the shedding profile of specific variants, age-specific shedding patterns, and possibly the impact of vaccination, all of which were not possible to explore during the time period of the submitted analysis. However, research within UKHSA has continued since submission to tackle these challenges, including

collaboration with colleagues in the WBE domain, and we plan to communicate these findings in the coming months. Further, we are examining the applicability of the approach to Scottish wastewater data, as part of research funding from the Turing Institute.

2) The paper also dives into the potential for using this system as an 'early warning' tool; however, there has been significant debate about this in the literature recently (some citations below). I would suggest revising these claims to include all appropriate considerations/caveats, and perhaps removing them from the abstraction

Olesen, Scott W., Maxim Imakaev, and Claire Duvallet. "Making Waves: Defining the lead time of wastewater-based epidemiology for COVID-19." *Water Research* (2021): 117433.
Bibby, Kyle, Aaron Bivins, Zhenyu Wu, and Devin North. "Making waves: Plausible lead time for wastewater based epidemiology as an early warning system for COVID-19." *Water Research* (2021): 117438.

Thank you for the references. In light of these we have added the following sentences to the discussion,

[Page 13, line 12] "This 4-day lead illustrates the potential of WW to be an early warning tool, even in an environment (such as England in 2020-2021) which had comprehensive clinical surveillance at the time. In circumstances with limited clinical surveillance, WW can provide accurate and timely estimates of regional prevalence with just a few samples."

3) Authors should include all relevant qPCR details (MIQE guidelines).

We thank the review(s) for making this very valid point. We have now included a description of the protocols used, making use of the MIQE guidelines. However, so as to not repeat the full protocols (published by Farkas et al. and Walker) we have provided a summary so an informed reader knows the processes involved and specifically the differences between protocols. These are now provided in the Methods and SI.

4) The validity of replacing non-detect values with weekly means is unclear. This assumes that non-detects were in error rather than a true result due to variation in the signal. This needs to be better justified or an alternative approach taken.

We apologise for confusion; insufficient details were provided to follow the methods. For clarification, the non-detect values were not replaced, and were fully utilised in the study. We have provided further clarification of how the raw data was analysed in the methods,

[page 18, line 8] "Of the total 6,228 samples supplied from the lab in the time-frame of this analysis, 1,365 samples returned a value of 'NA' for SARS-CoV-2, and were removed from the analysis, leaving 4,863 observations that were taken forward. Of these observations, 24.5% of replicate samples were below the limit of detection and 33.7% were below the limit of quantification; these values were retained in the analysis."

5) The authors cluster grab and composite samples but do not make a clear attempt to unify them. At a minimum I suggest discussion of the variability that this approach might take, or potentially an analysis of sample variability between these groups.

We have now expanded Table S1 to include summary data of the percentage of samples that were grab vs composite (54%). While exploring the percentage of samples that included replicates with non-detection, we did not identify any effect of the sampling method; in the exploration of whether sample type influences the probability of the non-detection no impact was detected ($p=0.365$). Further, a plot of the raw data for sites that had grab or composite did not show any substantial differences in SARS-CoV-2 when compared to the CIS positivity at the time, see plot below. We agree with the reviewer that this line of investigation is worth exploring (due to previous successes with composite sampling but their added costs), and is the subject of further investigation. We would be happy to correspond with the reviewer on this, if interested. The discussion of this is given below,

[page 15, line 19] "It is interesting to note that the percentage of samples from each WW site that were composite (as opposed to 'grab') had only a moderate effect on prevalence estimation. The use of composite samplers was dependent on their availability during the pandemic, with grab samples used for convenience as opposed to strategic intent. Ideally, this finding should be investigated using a comparative study design to investigate the possible added benefit of estimating SARS-CoV-2 from either approach."

REVIEWERS' COMMENTS

Reviewer #1 (Remarks to the Author):

The authors have clarified many of my concerns and have addressed all the substantive issues I raised previously. Very well done, and a great contribution to the science around wastewater surveillance for public health benefit. The study is extraordinary in that the authors actually have COVID-19 prevalence infection information through the rolling prevalence surveys. This article represents an enormous amount of work, and I congratulate and applaud the authors for those efforts. I have just two minor suggestions for terminology. The authors mentioned

1) Lines 41-42 in the abstract, I recommend sticking with "WW surveillance" rather than switching to "wastewater-based epidemiology" here. I would also suggest switching from "traditional surveillance" to "clinical surveillance", although as I write this I realize that the prevalence surveys perhaps are not quite "clinical surveillance". I'm concerned with the use of the term "traditional" here as wastewater surveillance and environmental surveillance more broadly has been used quite extensively for poliovirus and mosquito-borne illnesses, respectively.

2) Lines 56, 63, 75, 255, 261, 317 – I recommend using the term wastewater surveillance consistently throughout. Not sure about the UK but in the US there is a big debate about the terminology (monitoring v surveillance). Historically it has always been called surveillance, we call other things surveillance, and this should be no different. One acceptable alternative to wastewater surveillance would be infectious disease surveillance using wastewater. The switch to the term wastewater monitoring appears to have arisen from environmental engineers, environmental scientists, and microbiologists. The switch disenfranchises the fields of epidemiology and public health, for which this process has only one clear definition (surveillance). In the fields of epidemiology and public health the term monitoring means something else completely.

Reviewer #2 (Remarks to the Author):

The authors have adequately responded to my comments.

Reviewers comments

Reviewer #1 (Remarks to the Author):

The authors have clarified many of my concerns and have addressed all the substantive issues I raised previously. Very well done, and a great contribution to the science around wastewater surveillance for public health benefit. The study is extraordinary in that the authors actually have COVID-19 prevalence infection information through the rolling prevalence surveys. This article represents an enormous amount of work, and I congratulate and applaud the authors for those efforts.

Thank you for the kind words, it is truly appreciated!

I have just two minor suggestions for terminology. The authors mentioned
1) Lines 41-42 in the abstract, I recommend sticking with “WW surveillance” rather than switching to “wastewater-based epidemiology” here. I would also suggest switching from “traditional surveillance” to “clinical surveillance”, although as I write this I realize that the prevalence surveys perhaps are not quite “clinical surveillance”. I’m concerned with the use of the term “traditional” here as wastewater surveillance and environmental surveillance more broadly has been used quite extensively for poliovirus and mosquito-borne illnesses, respectively.

Throughout the document we have replaced “wastewater-based epidemiology” with WW surveillance, and opted to use the term “clinical surveillance”

2) Lines 56, 63, 75, 255, 261, 317 – I recommend using the term wastewater surveillance consistently throughout. Not sure about the UK but in the US there is a big debate about the terminology (monitoring v surveillance). Historically it has always been called surveillance, we call other things surveillance, and this should be no different. One acceptable alternative to wastewater surveillance would be infectious disease surveillance using wastewater. The switch to the term wastewater monitoring appears to have arisen from environmental engineers, environmental scientists, and microbiologists. The switch disenfranchises the fields of epidemiology and public health, for which this process has only one clear definition (surveillance). In the fields of epidemiology and public health the term monitoring means something else completely.

See above. We agree that terminology is challenging, further to your points above, veterinary epidemiologists use monitoring to refer to observing without implementing action.

Reviewer #2 (Remarks to the Author):

The authors have adequately responded to my comments.

Thank you for taking the time to review the manuscript.